# Interplay of biotic and abiotic factors shapes tree seedling growth and root-associated microbial communities
Joey Chamard[1,2,3,8], Maria Faticov [1,2,3,8] ✉, F. Guillaume Blanchet [1,4,5], Pierre-Luc Chagnon [6,7] & Isabelle Laforest-Lapointe [1,2,3] ✉

Root-associated microbes can alleviate plant abiotic stresses, thus potentially supporting adaptation to a changing climate or to novel environments during range expansion. While climate change is extending plant species fundamental niches northward, the distribution and colonization of mutualists (e.g., arbuscular mycorrhizal fungi) and pathogens may constrain plant growth and regeneration. Yet, the degree to which biotic and abiotic factors impact plant performance and associated microbial communities at the edge of their distribution remains unclear. Here, we use root microscopy, coupled with amplicon sequencing, to study bacterial, fungal, and mycorrhizal root-associated microbial communities from sugar maple seedlings distributed across two temperate-to-boreal elevational gradients in southern Québec, Canada. Our findings demonstrate that soil pH, soil Ca, and distance to sugar maple trees are key drivers of root-associated microbial communities, overshadowing the influence of elevation. Interestingly, changes in root fungal community composition mediate an indirect effect of soil pH on seedling growth, a pattern consistent at both sites. Overall, our findings highlight a complex role of biotic and abiotic factors in shaping tree-microbe interactions, which are in turn correlated with seedling growth. These findings have important ramifications for tree range expansion in response to shifting climatic niches.

Climate change, characterized by higher temperatures and increased frequency of extreme weather events, has major consequences for the spatial distribution of organisms[1]. Though it is now recognized that plant distributions are affected by changing climatic conditions[2–4] and that species tend to migrate poleward[5,6], there is accumulating evidence that non-climatic factors, such as biotic interactions, also play a critical role for plant colonization beyond their current geographic range[7–9]. This may partly explain why climatic conditions are often found to shift more rapidly than species range limits[10–12], especially for trees. A tree's ability to expand its geographic range in response to shifting climatic conditions relies on a complex combination of abiotic (e.g., variation in temperature, soil pH and nutrients, water availability) and biotic factors (e.g., pathogenic, competitive, mutualistic interactions). Research has shown that soil chemistry, such as pH[13], base cations[14,15], and nutrient availability[16], not only affect tree carbon

uptake and growth, but also shape the composition and functionality of root and soil microbial communities[17,18]. However, the interplay between abiotic and biotic factors in constraining or facilitating tree growth at range limits remains poorly understood, thus hindering our ability to predict tree species' range expansion in response to novel climatic conditions.

In the last twenty years, research on tree-microbe interactions has evidenced potential roles for soil microbes in driving tree recruitment and range expansion[19–21]. Most often such studies are restricted to narrow ecological guilds (with a particular emphasis on arbuscular mycorrhizal fungi; AMF), thus neglecting the complexity of tree-associated microbial communities[22–25]. Plant establishment and growth is influenced by a variety of microbial interactions, with root-microbe associations thought to be most influential for host fitness[26,27]. Beneficial root-associated microbes can promote plant growth[28,29], nutrient acquisition[30], and stress tolerance[31–34],

[1]Département de biologie, Université de Sherbrooke, Sherbrooke, QC, Canada. [2]Centre Sève, Département de Biologie, Université de Sherbrooke, Sherbrooke, QC, Canada. [3]Centre d'Étude de la Forêt, Université du Québec à Montréal, Montréal, QC, Canada. [4]Département de mathématiques, Université de Sherbrooke, Sherbrooke, QC, Canada. [5]Département des sciences de la santé communautaire, Université de Sherbrooke, Sherbrooke, QC, Canada. [6]Agriculture and Agri-food Canada, Saint-Jean-sur-Richelieu, QC, Canada. [7]Département des Sciences Biologiques, Université de Montréal, Montréal, QC, Canada. [8]These authors contributed equally: Joey Chamard, Maria Faticov. ✉e-mail: maria.faticov@gmail.com; isabelle.laforest.lapointe@gmail.com

while pathogenic microbes may cause disease[35,36]. These effects stem from interactions among different microbial guilds and kingdoms[17,37]. For example, non-mycorrhizal bacteria can either promote[38] or inhibit[39] symbiotic activity of AMF, with consequences on plant growth. In contrast, root and soil bacterial diversity were shown to be negatively affected by dark septate endophyte (DSE). DSE is a cryptic and diverse fungal group, whose interactions with plants range from mutualistic to pathogenic, impacting root and soil microbiomes as well as plant health[17]. Although more challenging, multi-kingdom and multi-guild microbial studies are essential to improve our understanding of microbial co-occurrence, interactions, and functions, based on their niche and resource use[40]. Studying tree-microbe relationships in forest ecosystems remains difficult because many microbes are still not well characterized taxonomically and ecologically. Yet, unraveling tree root microbial community assembly and variation *in natura* is crucial to better understand their role for tree species distribution in a changing climate.

Elevational gradients are frequently used to study plant-microbe interactions[41–43] based on the space-for-time substitution method[44]. This method is particularly useful to study ecological processes over short distance, allowing for the observation of novel biotic interactions. For example, the specificity of foliar fungal endophytes (microbes living within and among the cells of leaves) has been shown to peak at the core of a species distribution[41], where the host is most abundant. A lack of specificity has been demonstrated for root fungal endophytes at host range limits[45,46]. This lack of specificity could suggest that stochastic processes, such as dispersal and drift, play a greater role in driving microbial community assembly, thus partly explaining shifts in microbial community composition with elevation[43,47,48]. Yet, if biotic selection forces, driven by the plant host, are weaker at host range limits, abiotic selection forces (e.g., climatic conditions) could correspondingly increase in relative strength. The influence of abiotic (e.g., temperature, soil nutrients an moisture) and biotic factors (e.g., neighboring plant communities and microbe-microbe interactions) on root-associated microbial communities also varies across and within microbial kingdoms[49,50], thus complexifying our understanding of the dynamics of microbial communities. In this context, the field stands poised to integrate cutting-edge insights from (1) studies on tree-microbe interactions with (2) the refined application of established ecological theories to elucidate the mechanisms driving tree performance along elevational gradients.

Previously, tree species that successfully migrated beyond their current range limits have been shown to display less above- and belowground enemy pressures[51]. The *enemy-release hypothesis*, which posits that plant species experience reduced pathogen loads beyond their range limits[52], is often invoked to explain higher seedling density at species range limits[36], where propagule density is high and pathogen or predator densities are low[53]. More recently, several studies have added to this understanding, showing that for different tree species, variation in root bacterial and fungal community composition can be attributed to elevational and latitudinal gradients. This is explained by lower diversity of microbial communities, such as root bacterial endophytes and mycorrhizal fungi, at the edges of tree species ranges[54,55]. Additionally, greater tree regeneration has been observed at the upper elevation edge of their distribution, potentially due to a release from predation pressure[53]. On the other hand, tree colonization in non-native ranges could also be enhanced by novel associations with generalist mutualists[56], thus gaining greater benefits compared to their native range[57]. Such observations were previously theorized as the *enhanced mutualism hypothesis*[58,59], a theory that remains debated for AMF[25,60–62] and seems to be context-dependent[63]. Though these two hypotheses provide potential explanations for the role of tree-microbe interactions for plant distribution, to this day few studies have tested these theories at tree species range limits.

In this study, we investigated the drivers of tree seedling root-associated microbial communities and assessed their relationship with seedling growth along two elevation gradients. We sampled 100 sugar maple (*Acer saccharum*) seedlings along two elevation gradients in southern Québec, Canada. We characterized the microbial communities associated with each of these seedlings (AMF from both soil and roots, as well as root-associated bacteria and fungi) and measured soil chemistry, leaf nutrients, the composition of neighboring plant communities, canopy openness, and seedling growth. We predicted that (1) local soil chemistry and neighboring plant communities would be the main drivers of seedling root-associated microbial richness, diversity, and community composition; (2) seedling microbial richness, diversity, and community composition would shift across microbial kingdoms and elevation; (3) seedling growth would decrease with elevation because of shifts in microbial abundance and community composition (i.e., due to lower colonization by AMF at higher elevation or loss of symbionts) and unfavorable soil conditions (i.e., low pH and C, N, P, Mg, Ca, and K availability) closer to conifer-dense stands.

## Results

### Sugar maple seedling growth increases with elevation at both sites

Sampling was conducted along two elevational gradients located at Mont Écho near Sutton (45°6′46.09″N et 72°32′28.67″W; 811 m) and Mont Saint-Joseph in Mont Mégantic National Park (45°26′51″N, 71°06′52″W; 1075 m) in southern Québec, Canada (Fig. 1a; Supplementary Table 1). Both these gradients show steep shifts in tree community composition over short distances. For the remainder of this paper, we will refer to both locations as *Sutton* and *Mégantic*. The 98 seedlings in this study (Sutton [**S**] $n = 51$ / Mégantic [**M**] $n = 47$; Fig. 1a, b) were sampled along two elevation gradients of respectively 158 m and 120 m (Fig. 1c, d, Supplementary Table 1). At both sites, sugar maple seedling growth increased significantly with elevation (Fig. 2).

### Soil chemistry and conspecific metrics are key drivers of tree microbial communities

To identify the relative importance of elevation, canopy openness, soil chemistry, distance to conspecifics, conspecific diameter, AMF root length, arbuscule, vesicle colonization, and total DSE colonization, we performed linear regressions and PERMANOVAs (Fig. 3, Supplementary Data 1–5). In both sites, the same three predictors tended to best predict root fungal colonization: soil pH, soil Ca, and distance to conspecifics. In Sutton, soil Ca was the most common predictor for all microbial communities (Fig. 3a, Supplementary Data 1, 3, 5) but especially for root bacteria. Similarly, soil pH was an important driver of soil AMF, root fungi, as well as of root bacteria (Fig. 3a, Supplementary Data 1, 3, 5). On the other hand, distance to conspecifics showed a significant association with the diversity of soil AMF and root fungi (Fig. 3a, Supplementary Data 1, 3, 5). Finally, elevation only showed a significant relationship with root fungi (Fig. 3a, Supplementary Data 1, 3, 5). In Mégantic, soil pH was one of the most common predictors of variation for all microbial community alpha-diversity and community composition (Fig. 3b, Supplementary Data 2, 4, 5). Distance to conspecific also influenced microbial richness and community composition for soil and roots, showing for example a significant decrease in both root AMF and fungi richness, but not for bacteria. Soil Ca again predicted significantly root microbial community composition (Fig. 3b, Supplementary Data 2, 4, 5). In comparison, elevation was found to be a significant predictor of fungal richness and Shannon index as well as bacterial Shannon index, which all showed negative trends with elevation (Fig. 3b, Supplementary Data 2, 4, 5, Supplementary Fig. 1). Conspecific diameter was a significant predictor mostly for soil AMF. Finally, vesicle and DSE colonization showed a significant association with soil and root AMF community composition (Fig. 3b, Supplementary Data 2, 4, 5).

### Strong covariation between abiotic and biotic conditions with root microbial communities

At Sutton, both root and soil AMF covaried significantly with the aboveground environment (RVs of 13.5%, $p = 0.025$ and 17.3%, $p = 0.025$, respectively), conspecific and root microscopy metrics (RVs of 21.7%, $p = 0.002$ and 9.7%, $p = 0.016$), soil chemistry (RVs of 16.3%, $p = 0.005$ and 15.4%, $p = 0.023$), but not with host characteristics or neighboring plant

Fig. 1 | Sampling design along two elevation gradients. a White squares indicate the location of seedlings along elevation gradients respectively in Sutton and Mégantic, both located in Southern Québec, Canada. Shades of gray indicate bioclimatic domains. b Schematic of performed analyses with the various seedling tissues. c, d Seedling location at each elevation gradient displayed in meters above sea level (masl) overlaid on gray relief. Starting at 567 and 675 masl (Sutton and Mégantic), seedlings were randomly sampled until species distribution limit at ~725 and ~796 masl respectively. Projection: NAD 1983 MTM 8 (Sutton) & MTM 7 (Mégantic). Sources: Québec Ministère des Ressources Naturelles et des forêts (2005) and Ministère des Forêts, de la Faune et des Parcs (2020). Author: Center for Forest Research 2022.

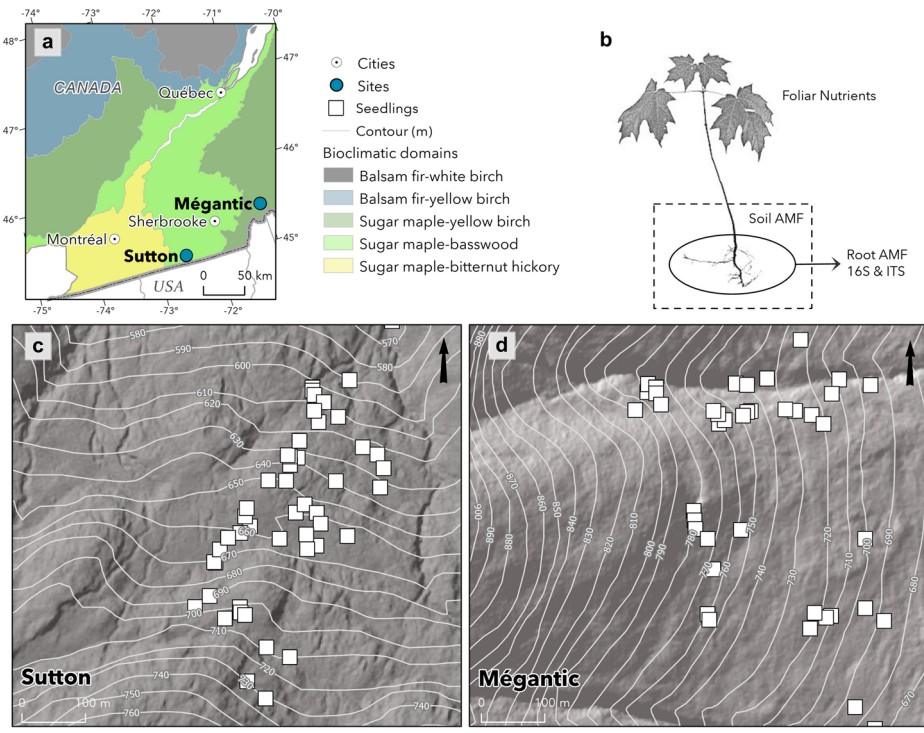

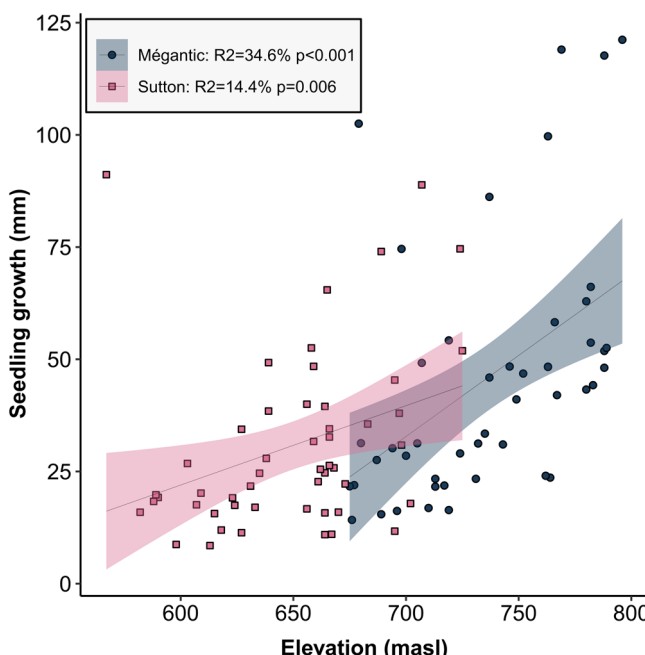

Fig. 2 | Seedling growth significantly increases with elevation. Point colors indicate site identity (blue for Mégantic and pink for Sutton). Line shading represents 95% intervals from smoothed conditional means with a linear model regression. The legend shows statistical significance and coefficient of determination ($R^2$) per site based on a linear regression model. $N$ = 51 and 47 for Sutton and Mégantic samples, respectively.

community (Supplementary Table 2, Fig. 4a). Additionally, fungal, and bacterial communities demonstrated significant correlations with environmental factors. The fungal community was strongly linked to the aboveground environment (RV 24.7%, $p < 0.001$; Supplementary Table 2,

Fig. 4b), while the bacterial community showed a significant association with soil chemistry (RV 33.8%, $p < 0.001$; Supplementary Table 2a, Fig. 4b).

In Mégantic, the root AMF community covaried significantly with the aboveground environment (RV 14%, $p = 0.014$) as well as with conspecific and root microscopy (RV 8.6%, $p = 0.001$), but not with host characteristics, neighboring plant community or soil chemistry (Supplementary Table 2, Fig. 4c). The soil AMF community, however, did correlate significantly with host characteristics (RV 23.9%, $p = 0.003$) and the neighboring plant community (RV 32%, $p = 0.005$; Supplementary Table 2c, Fig. 4c). Furthermore, both fungal and bacterial communities were found to significantly correlate with all environmental matrices, but with varying strengths of association. The highest correlations were found between fungi and the neighboring plant community (RV 41.7%, $p < 0.001$) and between bacteria and the aboveground environment (RV 44.6%, $p < 0.001$; Supplementary Table 2c, Fig. 4d). To summarise, root AMF correlates significantly with the aboveground environment and root microscopy, but not with host or soil chemistry at Mégantic.

Across both sites, while all root microbial matrices showed correlations with each other, there were a few exceptions, such as between bacterial and soil AMF communities in Sutton (Supplementary Table 2b). The highest correlations were found between bacterial and fungal communities (RVs of S: 82.3% and M: 86.3%, $p < 0.001$; Supplementary Table 2d), while the lowest between bacterial and root AMF at Sutton (RV 50.4%, $p < 0.001$; Supplementary Table 2d), as well as between root and soil AMF (RV 23.2%, $p = 0.013$; Supplementary Table 2d) at Mégantic.

### Soil pH and root fungi influence seedling growth

In Sutton, our results revealed that soil Ca (std.coef [SC] = 0.32, $p = 0.024$), soil AMF MDS1 (SC = 0.56, $p < 0.001$), and fungal MDS1 (SC = 0.61, $p < 0.001$) were key drivers of seedling growth, collectively explaining 41% of its variation (AIC = 301, Fisher's C = 12.98, $p = 0.528$; Fig. 5a). Soil pH and AMF vesicle colonization were found to significantly influence soil AMF MDS1, accounting for 19% of its variation (SC = −0.27, $p = 0.045$ and SC = −0.36, $p = 0.007$, respectively). Additionally, soil pH (SC = 0.59, $p < 0.001$) and Ca (SC = -0.67, $p < 0.001$) were the primary factors explaining 67% of the variation in fungal community composition (Fig. 5a).

**Fig. 3 | Environmental factors drive sugar maple root microbial endophyte alpha- and beta-diversity, as well as community composition.** Predictors (**a** Mégantic; **b** Sutton) are ordered based on how frequently they showed significant associations in final models (after selection). Numbers indicate standardized coefficients (linear models) or sums of squares (PERMANOVAs for composition). Diagonal stripes show lines without significant drivers. $N = 51$ and 47 for Sutton and Mégantic samples, respectively. For test statistic details, degrees of freedom, and $R^2$ see Supplementary Data 1–5. *$p ≤ 0.05$, **$p ≤ 0.01$, ***$p ≤ 0.001$.

**a**

| Responses | | Soil Ca | Soil pH | Distance to conspecific | Seedling growth | Soil P | Elevation | DSE colo. | Soil moisture | Arbuscule colo. | Vesicle colo. |
|---|---|---|---|---|---|---|---|---|---|---|---|
| AMF Root | Richness | | | | 5.20* | | | | | | |
| | Shannon index | | | | | | | | | | |
| | MDS1 | | | | | | | | | | |
| | MDS2 | 0.42** | | | | | | -0.01** | | | |
| | Composition | 0.56** | | | | | | | | | |
| AMF Soil | Richness | | | | | | | | | | |
| | Shannon index | 0.18* | | 0.12* | | | | | | | |
| | MDS1 | | | | 0.26* | | | | | | -0.24* |
| | MDS2 | | -0.72** | | 0.17* | | | 0.01* | -0.29** | | |
| | Composition | 0.44* | | | | | | | | | |
| Fungi | Richness | | 20.24* | | | | | | -7.19* | | |
| | Shannon index | | | 0.20* | | | | | | | |
| | MDS1 | -0.47*** | 0.68** | | | | | | | | |
| | MDS2 | | | | 0.19* | | 0.01** | - | | | |
| | Composition | 0.56*** | 0.54*** | | | | 0.47** | | | 0.39* | |
| Bacteria | Richness | 51.28* | | | -30.11* | | | | | | |
| | Shannon index | 0.24** | | | | | | | | | |
| | MDS1 | -0.45*** | 1.25*** | | | | | | | | |
| | MDS2 | -0.49*** | -0.73*** | | | | | | | | |
| | Composition | 0.59** | 0.89*** | | | | | | | | |

**b**

| Responses | | Soil pH | Distance to conspecific | Soil Ca | Elevation | Conspecific diameter | Vesicle colo. | Seedling growth | DSE colo. |
|---|---|---|---|---|---|---|---|---|---|
| AMF Root | Richness | | -6.1** | | | | | | |
| | Shannon index | | | | | | | | |
| | MDS1 | 1.17*** | | | | | | | |
| | MDS2 | -0.65** | | 0.31* | | | | | |
| | Composition | 0.56** | 0.51** | | | | 0.36* | | 0.38 |
| AMF Soil | Richness | | | | | 6.19* | | | |
| | Shannon index | | | | | 0.10* | | | |
| | MDS1 | 0.48* | 0.27* | | | -0.20* | | | |
| | MDS2 | 0.55* | | | | | | | |
| | Composition | 1.26** | 0.71*** | 0.56* | | | | | |
| Fungi | Richness | | -7.01* | | -0.21* | | | | |
| | Shannon index | | -0.26** | | -0.01* | | | | |
| | MDS1 | -1.27*** | | | | | | 0.23** | |
| | MDS2 | | -0.39** | | | | | | |
| | Composition | 1.09*** | 0.49** | 0.48** | 0.40* | | | | |
| Bacteria | Richness | | | | | | | | |
| | Shannon index | -0.26** | | | -0.01*** | | | | |
| | MDS1 | 1.17*** | | | | | | | |
| | MDS2 | -0.65** | | 0.31* | | | | | |
| | Composition | 1.32*** | | 0.44** | | | | | |

In Mégantic, soil pH (SC = 0.57 $p = 0.009$) and elevation (positive effect, SC = 0.49, $p < 0.001$) explained 52% of the variation in growth, thus being the main factors influencing this variable (AIC = 254, Fisher's C = 14.94, $p = 0.529$; Fig. 5b). Fungal community composition, represented by fungal MDS1, was primarily affected by soil pH (SC = −0.79, $p < 0.001$) and distance to conspecific (SC = 0.17, $p = 0.043$), which together explained 72% of its variation along the elevation gradient. This finding suggests that changes in fungal composition could indirectly affect seedling growth through soil pH and distance to conspecific (Fig. 5b). Furthermore, root AMF MDS1 was significantly associated with distance to conspecific (SC = 0.38, $p = 0.004$), soil pH (SC = 0.35, $p = 0.009$), and root DSE colonization (SC = −0.32, $p = 0.016$), explaining 37% of the variation in root AMF composition; yet it did not show a significant direct association with seedling growth.

**Fungal functional guilds and fungal taxa associated with seedling growth**

In total, we assigned functional guilds to 1018 and 1163 root fungal ASVs in Sutton and Mégantic, respectively. In Sutton, out of 1018 ASVs, 19 (2%) were other fungi, 38 (4%) pathogens, 167 (16%) mutualists, 309 (30%) saprotrophs, and 484 (48%) unknown; in Mégantic, of 1163 ASVs, 19 (2%) were other, 54 (5%) pathogens, 86 (7%) mutualists, 355 (30%) saprotrophs, and 649 (56%) unknown (Supplementary Data 6, 7, respectively). None of the fungal functional guilds differed in their relative abundances with seedling annual growth in both sites (Supplementary Table 3). Two fungal families (decrease in *Dermateaceae* and *Chaetosphaeriaceae*; Supplementary Table 4a, b) and two genera (increase in *Gyoerffyella* and decrease in *Pezicula*; Supplementary Table 4c, d) differed in the relative abundance with higher seedling growth at Sutton. Interestingly, the genus *Gyoerffyella* was assigned to fungal saprotrophs, while *Pezicula* to root endophytes (see Supplementary Data 6).

**Discussion**

As climate change alters plant species distribution, a growing body of evidence suggests that both climatic and non-climatic factors (including the interactions between plants and soil microbes) play pivotal roles in shaping plant range expansion[64,65]. Our study reveals that a complex interplay of abiotic and biotic factors drives sugar maple (*Acer saccharum*) tree root-

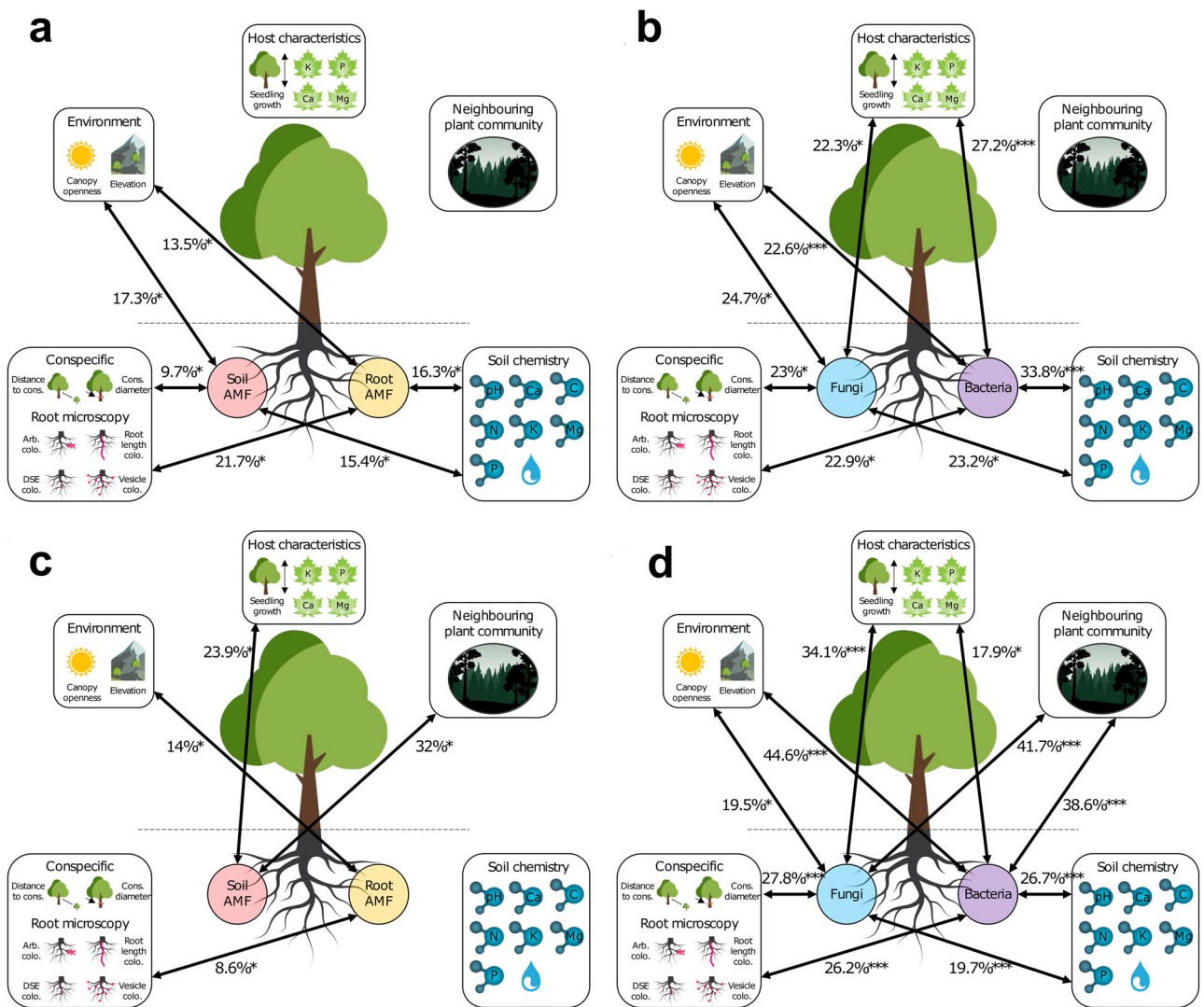

**Fig. 4 | Multivariate correlations between the four root microbial community matrices and (i) the environment, (ii) host characteristics, (iii) neighboring plant community, (iv) conspecific and root microscopy metrics, as well as (v) soil chemistry.** Multivariate correlations for (**a** and **c**) root and soil AMF; as well as for (**b** and **d**) fungal and bacterial communities in (**a** and **b**) Sutton and (**c** and **d**)

Mégantic. Lines indicate significant correlations (see Supplementary Fig. 6 for insignificant correlations). Numbers show RV coefficients (degree of association between matrices in percentages). For all RV coefficients and significance statistics see Supplementary Table 2. *$p \leq 0.05$, **$p \leq 0.01$, ***$p \leq 0.001$.

associated microbial communities, which in turn impact seedling growth. Soil pH, soil Ca, and distance to conspecific are shown to be important drivers of root-associated microbial alpha-diversity and community composition (Fig. 1). The direct and indirect effects of these abiotic and biotic drivers on seedling growth further demonstrate how environmental factors (soil pH, Ca, distance to conspecific) and microbial community composition (root fungi, soil AMF, root DSE colonization) can impact the performance and potentially tree species adaptation to changing environmental conditions (Fig. 5). Overall, this study provides insights into the complex interactions between trees and root-associated microbes spanning multiple microbial kingdoms, and how they affect tree growth, particularly at the range limits. These findings emphasise the need to improve our understanding of the role of tree-microbe and microbe-microbe (e.g., bacteria-fungi) interactions *in natura* to reveal their influence on tree species distribution and migration in changing climatic conditions.

Soil pH was a key driver of community composition across microbial kingdoms, in line with previous studies which established pH as an important edaphic factor influencing microbial communities along both latitudinal and elevational gradients[66–69]. Soil Ca, which plays key functions for microbes by enhancing microbial cell wall stability, nutrient uptake, and

pH buffering, also impacted significantly soil and root endophytic richness, diversity, and community composition[70]. Interestingly, the importance of distance to conspecific on root AMF, soil AMF, and root fungi could be explained by community assembly mechanisms such as dispersal or selection (e.g., plant-soil feedback). Dispersal limitation could lead to soil and root microbial community turnover as distance to conspecific trees increases, thus explaining in part the observed decrease in root AMF and fungal diversity with higher distance to conspecific trees in one of the sites[71]. On the other hand, plant-soil feedback may influence root and soil microbial richness, diversity, and composition as trees can alter the soil, predominantly through the secretion of specific root exudates to recruit or repel certain microbial taxa[72,73]. Over time, these biochemical processes lead to the selection of distinct microbes in the soil surrounding conspecific trees. As distance from conspecific trees increases, the influence of conspecific root exudates is reduced, leading to a turnover in microbial communities, even when environmental conditions remain constant[74].

In our study, DSE colonization showed a significant association with root and soil AMF at both sites. However, the interpretation of this association, as positive or negative, is not straightforward, particularly due to the complexity of interpreting data from axis scores (e.g., MDS), which mainly

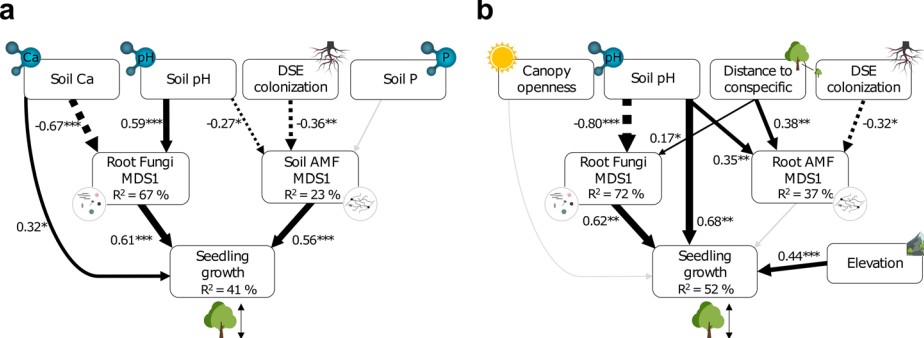

**Fig. 5 | Structural equation models (SEM) representing the relationships among environmental factors, root microbial communities, and seedling growth.**
**a** Most parsimonious SEM showing relationships between seedling growth, environmental variables, and root microbial community composition (soil AMF and root fungal MDS1) for Sutton. **b** Most parsimonious SEM showing relationships between seedling growth, environmental variables, and root microbial community composition (root AMF and fungal MDS1) for Mégantic. Solid and dotted black arrows indicate significant positive and negative paths, respectively, while gray arrows represent non-significant paths. Conditional $R^2$ values (%) are displayed below corresponding exogenous variables. Arrow width indicates the strength of the standardized coefficients. *$p \leq 0.05$, **$p \leq 0.01$, ***$p \leq 0.001$.

capture the broad variation in soil and root AMF community composition rather than direction of the interactions. This ambiguity is further compounded by the diverse and understudied ecological roles of DSE, which can range from beneficial to detrimental[75,76]. If some DSE are indeed mutualistic and positively associated with certain AMF taxa, such tripartite interactions between these fungal groups and plant roots might enhance the overall health and performance of the plant host. However, considering the diversity within DSE, encompassing various taxa from several orders of the phylum Ascomycota, different DSE might show contrasting interactions with plant roots and AMF[17]. On the other hand, at least some DSE could also be detrimental to *Acer*, and the relationships we observe in our study might be antagonistic. In particular, the order *Pleosporales*, which was found at higher relative abundance in sugar maple by De Bellis et al.[75], might play a crucial role in these interactions. Interestingly, a recent study by Netherway et al.[17] reported an effect of DSE colonization on root and soil fungi and bacteria community composition and functional genes, highlighting an intriguing role of DSE in structuring microbial interactions belowground. Future work combining fieldwork and controlled experiments will be instrumental to understand the functional roles of DSE on microbe-microbe and tree-microbe interactions[17,75,76].

Contrary to our predictions, elevation was not a primary driver of the shifts in microbial communities at both sites. However, we did observe a shift in fungal community composition along elevation at Sutton. Similarly, we detected a slight decrease in fungal richness and Shannon diversity as well as a decrease in bacterial richness and a hump-shaped relationship of root bacterial diversity and elevation at Mégantic (Supplementary Fig. 1d, f, h). Notably, root and soil microbial richness and diversity responses to elevation show different patterns across studies: some indicate an increase in soil fungi diversity with elevation[77], while others report a decrease[78]. Additionally, certain studies have identified a hump-shaped[79], U-shaped[54,80] or no change[81] in diversity patterns along elevational gradients. As for sugar maple microbiome, a previous study investigating changes in microbial richness, diversity, and community composition along the same elevational gradient at Mégantic, but using only two elevations (within and at species range edge), had documented a decrease in root bacterial diversity[54]. Interestingly, our dataset shows a hump-shaped relationship between root bacterial diversity and elevation at the same site (Supplementary Fig. 1d), thus providing more resolution on this spatial pattern and warranting future research into which abiotic or biotic factors contribute to this relationship at mid-elevational ranges.

Covariation analyses demonstrated that root AMF communities were correlated with the aboveground environment (canopy openness and elevation), conspecific and root microscopy metrics, and soil chemistry (only in Mégantic), but not with host characteristics. The microbial communities that showed a significant correlation with host characteristics were only

fungi and bacteria in Mégantic. This correlation can be attributed to the well-established relationship between host nutritional status and community composition[82], as well as the influence of microbial metabolites on host nutrient uptake and growth[83,84]. In contrast to our predictions, neighboring plant community had a weak covariation with root and soil AMF communities. Similarly, while fungi and bacteria covaried with almost all studied environmental matrices, they showed no covariation with neighboring plant community in Sutton[85–87]. To better understand codependency between AMF and neighboring plant communities, future experimental studies could try to keep constant the plant or AMF community while altering the other. These multivariate correlations support our earlier results showing the importance of distance to conspecific for sugar maple root-associated microbes. Notably, comparatively weak correlations of neighboring plant community and higher correlations of distance to conspecific with root-associated microbes are in line with a growing number of studies that demonstrate that understory plant communities have a minor contribution to microbial communities as compared to dominant canopy trees[88,89]. For example, dominant canopy trees, with their extensive root systems and interactions with AMF and ECM fungi, are more likely to have a pronounced impact on soil nutrient profiles and in return influence root microbial communities.

Very rarely have studies on plant-microbe interactions provided simultaneous data on soil mycorrhizae, root mycorrhizae, bacteria, and fungi with both microscopy and genomics. This strength of our study allowed us to highlight the significant correlations across almost all root-associated microbial communities. These results suggest that, not only higher-level processes such as dispersal limits or host selection can alter microbial community composition, but that strong and complex inter-kingdom interactions occur and contribute to shaping these communities. For example, the strong correlation observed between bacterial and fungal communities (S: 82%, M: 86%), suggests that these two microbial kingdoms are deeply linked by trophic interactions (e.g., certain fungal groups may use bacterial metabolites) and/or respond to largely similar environmental filters[90,91]. On the other hand, a weaker correlation between root and soil AMF (23%), at least in one of the two sites, suggests niche differentiation of soil and root AMF species, plant selectivity towards certain AMF species or strains, or differences in resource allocation between soil and roots[92,93].

In contrast to what we initially predicted, our data showed a positive relationship between seedling growth and elevation at both sites (Fig. 2). In an *in natura* study design in which we could not control for external variables, our SEM models explained respectively 42% and 52% of seedling growth at both sites. These models highlighted the preponderant effect of soil chemistry and root fungi, but not canopy openness or neighboring understory plant communities, in driving sugar maple growth along two elevation gradients. Of note, many studies have previously showed a positive

relationship between canopy openness (in particular canopy gaps) with seedling growth, an effect that varied with the identity of the neighboring canopy tree species[94,95].

When exploring which fungal taxa may explain the observed pattern, we found that the genus *Gyoerffyella* (a fungal saprotroph) increased, while the genus *Pezicula* (a root endophyte) decreased in their relative abundances with higher seedling growth in Sutton. At this stage it is unclear how changes in the relative abundance of these two genera are linked to seedling performance, warranting targeted experiments to understand the underlying mechanisms. The higher seedling growth at the edge of sugar maple's distribution could be explained by the *enemy-release* hypothesis, a pattern previously observed at Mégantic[53]. However, in our study, we did not find any evidence of fungal pathogens (or any other functional groups) changing in their relative abundance with seedling annual growth. This could suggest that there are no real hotspots of root pathogens, or higher AMF or ectomycorrhizal abundances along seedling growth at both sites. Similarly, we did not find evidence that would support the *enhanced mutualism hypothesis*. For example, neither AMF colonization nor relative abundance showed a significant association with seedling growth. These findings indicate that other aspects of fungal communities, for example network complexity and inter-taxa or inter-kingdom interactions, rather than their relative abundances or colonization rates, could have significant influence on seedling performance. Notably, the higher seedling growth at the edge of distribution could also be explained by other mechanisms, such as lower herbivore pressures or microclimatic variation (e.g., temperature) along the gradient.

Understanding the mechanisms through which fungal community influences seedling growth proves to be even more challenging when considering the strong covariation between fungal and bacterial community matrices in our study ($RV_{Sutton}$: 82.3% and $RV_{Mégantic}$: 86.3%). This strong correlation suggests that bacterial and fungal communities may not simply coexist, but also interact in a manner that impacts plant growth[96,97]. Importantly, trees not only respond to, but also shape their root and soil microbial communities (e.g., by recruiting or repelling certain microbial taxa with root exudates[98] or by changing microbial legacy in the soil[99]). Thus, to fully understand plant-microbe relationships and their implications for plant performance and adaptation, future research should not only focus on experimental manipulations decoupling the impacts of inter-kingdom microbial interactions, but also investigate the role of trees in shaping belowground microbial communities.

In conclusion, our results demonstrate that soil chemistry (pH and Ca) as well as distance to the closest conspecific trees are key drivers of root-associated microbial communities. Sugar maple root microbial communities covary strongly across elevation gradients, especially root bacteria and fungi. As sugar maple seedling growth increased with elevation, soil pH and root fungal community composition were correlated with seedling growth. Altogether, our data provides evidence of the importance of both biotic and abiotic factors on tree-microbe interactions and tree host growth. Our work is one of the most comprehensive studies of tree-microbe interactions *in natura* and suggests several major avenues to investigate further the roles of tree root microbiota at species range limits in the context of climate change.

## Methods
### Study sites
From 1981 to 2010, the mean annual temperature was 6.1 °C (Sutton, value measured ~238 masl) and 4.0 °C (Mégantic, value measured ~240 masl), and the mean annual total precipitation was ~1310 mm and ~1370 mm, measured at the nearest weather station for Sutton and Mégantic, respectively[100]. As for soil properties, the upper layer of soil tends to be of high acidity (pH of 4.6 ± 0.41 and 4.6 ± 0.5)0 in Sutton and Mégantic; and high carbon-to-nitrogen (C:N) ratios (Supplementary Table 1)[8]. Along these elevational gradients, variations in abiotic conditions are portrayed by a transition of vegetation from AMF-dominated stands of temperate tree species (mostly sugar maple [*Acer saccharum*], with some companion ectomycorrhizal [ECM] species such as American beech [*Fagus grandifolia*]

and yellow birch [*Betula alleghaniensis*]), to ECM-dominated stands of boreal tree species including balsam fir (*Abies balsamea*) and spruces (*Picea* spp.).

In summary, when comparing the two gradients (Supplementary Table 1), Mégantic showed a higher elevation ($p < 0.001$), canopy openness ($p = 0.011$), exchangeable soil calcium ($p < 0.001$), and magnesium ($p < 0.001$), seedling growth ($p = 0.001$), as well as foliar phosphorus ($p = 0.001$) and potassium ($p = 0.001$). Sutton showed a higher soil moisture ($p = 0.041$), total root length colonization by AMF ($p = 0.016$), and colonization by DSE ($p < 0.001$; Supplementary Table 1).

To characterize sugar maple root-associated fungi, bacteria, and AMF along the elevational gradients, in 2021, we randomly sampled 50 sugar maple seedlings ($\bar{x} = 12.3$ years ± 4.3 standard deviation (SD)) at each site in stands that had not been managed for at least 60 years (Fig. 1b–d). One extra seedling was sampled at Sutton and three were lost at Mégantic, yielding a total of 98. For each seedling, we measured mean annual tip growth (terminal internode length mm), age (years), height (cm), and root collar diameter (mm) as well as canopy openness (using 360° photos above each seedling using a Gap light analyzer). To capture understory plant communities in early season, for each plant species within 1 m radius of each seedling we estimated percent cover using the following five classes: (1) 0%–understory plants are absent; (2) 1%–25%, (3) 25%–50%, (4) 50%–75%, and (5) 75%–100%. We also measured the distance to the closest adult conspecific tree (DBH > 10 cm) and its diameter at breast height. We collected seedling leaves, roots, and surrounding soil for chemical and molecular analyses. Soil samples (20 cm-deep cores, matching depth of root systems of seedlings) were used to characterize AMF communities and estimate soil properties (see below), while root samples were used to characterize root-associated AMF, bacteria, and fungi (see details below). Soil and root samples were kept on ice upon transportation to the laboratory, where they were stored at −20 °C (on the day of collection).

### Root sample preparation
Roots were cleaned using distilled water to remove residual attached soil particles and organic matter. They were then divided into two subsamples, one for microscopy and the other for molecular analyses. Subsamples for mycorrhizal colonization were stored in FAA solution (5:5:50:40 formaldehyde: glacial acetic acid: 95% ethanol: distilled water) until root clearing-staining. The second portion of acquisition roots was surface-sterilized as in Wallace et al.[54] to remove epiphytic organisms. Roots were submerged in 15 mL of 70% ethanol and vortexed at high speed for five minutes, then vortex-rinsed three times with DNA-free water for three minutes. Roots were then stored in sterile 2 mL tubes at −80 °C.

### Soil nutrients analysis
To quantify soil nutrients and moisture content, soil samples were first homogenized. Then, 2 mL per sample were transferred to a sterile tube for molecular analyses. The remaining soil was placed in a 1 L aluminum foil and oven-dried at 65 °C for 48 h to determine moisture content in the soil (g water per g dry soil). Samples were then sieved (0.5 cm) to remove rocks and debris. 2 g of soil were used to determine pH in a 1:2 soil:water volumetric ratio[101]. A second subsample of 10 mL was finely ground using a soil mill (Ball Mill). Subsequently, physicochemical analyses were carried out on two soil subsamples per seedling (total = 196 samples). Total carbon and nitrogen were determined using dry combustion (LECO CR-412, LECO Corporation, St. Joseph, MI, USA), while phosphorus and base cations (calcium, magnesium, and potassium) were extracted from 2.5 g dry soil with 25 mL of Mehlich-III solution and quantified using inductively coupled plasma emission spectrophotometry (ICP-AES).

### Root staining and mycorrhizal quantification
To quantify mycorrhizal and dark septate endophytes (DSE) colonization, we first stained sugar maple roots using the protocol of Vierheilig et al.[102] (Supplementary Fig. 2). In summary, samples were treated for 4 h with 10%

https://doi.org/10.1038/s42003-024-06042-7                                                                    **Article**

KOH at 90 °C in a water bath and then exposed to an alkaline hydrogen peroxide solution (15% NH₄OH, 15% H₂O₂; 70% H₂O) for 15 min. When roots showed little root discoloration after these two steps, they were exposed a second time to 10% KOH at 90ºC in a water bath but this time with checks every 15 min to limit the alteration of the acquisition roots. When roots had lost their epidermis and showed a golden coloration (Supplementary Fig. 2), we performed a rinse with distilled water followed by a five-minute soak in a 1% acetic acid solution at room temperature. Roots were then exposed for 4 min in a 5% v/v solution of ink (Waterman Mysterious Blue) in 5% acetic acid. Successive washes in distilled water were performed to remove excess ink. Samples were stored in 50% lactoglycerol for a minimum of 24 h before slide observation under light microscopy to remove excess dye.

Stained root samples were then mounted on slides using a semi-permanent mounting solution[103] (Supplementary Fig. 2). Root length colonization by arbuscules, vesicles, DSE, were scored using the grid-line intersect method[104]. DSE are ubiquitous ascomycete fungal root colonizers grouped based on morphological characteristics[105,106]. Although to this day the effect of DSE on plant hosts is unclear, they can act as pathogens and their virulence has been correlated to root colonization magnitude[107].

## Foliar elements quantification

Leaf samples were used to measure foliar elemental concentrations (K, P, Ca, and Mg) according to the protocol of Renaudin et al.[75]. Samples were air-dried for 72 h and grounded in liquid N2. Fifty milligrams of air-dried sample were digested in 2 mL of nitric acid (trace metal-free grade, ThermoFisher Scientific) and 200 μl of hydrogen peroxide (trace metal-free grade, MilliporeSigma). Digestions started at room temperature for 30 min, followed by 1 h at 45 °C and 2 h at 65 °C in a heating block digestion system (DigiPREP, SCP Sciences). Digested samples were diluted with Milli-Q water (MilliporeSigma) to reach a 2% v/v acid concentration. Ca, Mg, K, and P concentrations were measured by ICP-MS (X-Series II, ThermoFisher Scientific) using rhodium (Rh) as the internal standard. All leaf nutrient analyses were carried out in triplicates and concentrations were reported as ppm (μg of element per g of leaf dry weight). Two samples could not yield foliar nutrients measurements.

## Soil and root DNA extraction, amplification, and sequencing

DNA was extracted from approximately 150 mg of fresh roots using the PowerSoil DNA Isolation Kit (QIAGEN, Hilden, Germany) with two modifications following De Bellis et al.[75]. Soil DNA was extracted from 250 mg of homogenized soil with the same extraction kit following the manufacturer's protocol. All extracted DNA extracts were then stored at −20 °C.

To characterize root bacterial communities, we amplified the V5-V6 region of the bacterial 16S ribosomal RNA gene using the chloroplast-excluding primers 799F-1115R[108,109]. For root fungal communities, the internal transcribed spacer (ITS) region was amplified using fungal-specific primers ITS-1F and ITS2[110,111]. For root and soil AMF communities, we used the Glomeromycota-specific primer AML2[112] and the universal eukaryotic primer WANDA[113]. Primer sequences and reaction conditions for each microbial group are showed in Supplementary Table 5. For all amplicons, PCR products were visualized on 2% agarose gel and were normalized using Just-a-plate 96 PCR purification and normalization kit (CharmBiotech) following the manufacturer protocol. Multiplexed amplicon libraries for each of the three groups were prepared by mixing equimolar concentrations of DNA. Pools were purified with AMPure XP using the manufacturer protocol (Beckman Coulter). Quality control of the libraries as follows: libraries were quantified using the Qubit™ dsDNA HS Assay Kit (Invitrogen™) and the NEBNext® Library Quant Kit for Illumina® (New England BioLabs). Average size fragment was determined with Bioanalyzer (Agilent). Before sequencing, PhiX control library (Illumina) was spiked into the amplicon pool to improve the unbalanced base composition. Sequencing was performed on Illumina MISEQ (CERMO-FC platform, UQÀM).

## Bioinformatics

Read processing was conducted using *dada2*[114] in R[115] including quality and chimera filtering. Bacteria and root fungi read processing was conducted using *dada2* in R, which included quality and chimera filtering. Specifically, for bacteria, the processing involved setting maximum expected errors (argument *maxEE*) to a threshold of 2 for both forward and reverse reads, truncating reads at a quality score below 2 (argument *truncQ*), and truncating forward reads to 260 bp and reverse reads to 200 bp (argument *truncLen*). Additionally, the first 19 bases of forward reads and the first 16 bases of reverse reads were trimmed (argument *trimLeft*) to remove primer sequences and low-quality starting bases. For fungi, the processing involved setting maximum expected errors (argument *maxEE*) to a threshold of 2 for both forward and reverse reads, truncating reads at a quality score below 2 (argument *truncQ*) and retained sequences of minimum 50 base pairs or longer for the subsequent steps in the analysis. For bacteria and root fungi, to reduce the presence of bioinformatic artifacts, we filtered amplicon sequence variants (ASVs) with <10 reads in their respective dataset, as well as ASVs not identified at the kingdom level.

16S taxonomy was assigned with SILVA version 138.1[116] and for ITS with UNITE version 8.3[66]. For root and soil AMF, taxonomy was assigned using an evolutionary placement algorithm (EPA)[67]. To affiliate reads via EPA, we first filtered the ASV table to keep: (1) ASVs that were present in more than one sample and (2) ASVs that, after rarefaction of the samples at 4000 reads, still had more than 100 reads. We then ran a series of BLAST analyses against NCBI (SSU_eukaryote_rRNA and 18S_fungal_sequences) and MaarjAM databases to remove ASVs that scored high against non-AMF eukaryotes and other fungi, but not high against known AMF sequences, or simply ASVs that were nowhere close to sequences found in MaarjAM[68]. Then we aligned the filtered ASVs with the 1.5 kb reference sequence published by Krüger et al.[69]. The maximum-likelihood phylogenetic backbone tree for these reference sequences was calculated using RAxML[117] and epa-ng[118] was used to map query sequences on the reference tree. Finally, we used GAPPA (a command line interface for phylogenetic placement analysis)[119] to assign the most likely taxonomy to each query ASV.

For bacteria, we obtained a final dataset of 2,356,945 sequences assigned to 7066 ASVs (Supplementary Table 6). For fungi, we obtained a final dataset of 1,745,695 sequences assigned to 1797 ASVs (Supplementary Table 6). Finally, for root and soil AMF respectively, we obtained a final dataset of 344,267 and 95,011 sequences assigned to 182 and 173 ASVs (Supplementary Table 6). Summary statistics for (i) quality, chimera-filtered sequences, and average sequence size, (ii) final ASV counts as well as (iii) taxonomical annotation of the four final datasets are respectively presented in Supplementary Tables 6 and 7.

In total, for root and soil AMF, we rarified each sample to 4000 reads, when assigning taxonomy. For bacteria and fungi, the resulting read counts after rarefaction were the following: (i) for bacteria, we rarified to 9000 reads for Mégantic and 12,000 reads for Sutton; (ii) for fungi, dataset was rarified to 5500 reads for Mégantic and 4000 for Sutton.

To understand the relationship between fungal guilds, seedling annual growth, and elevation gradients, we manually assigned functional traits to fungal ASVs based on their taxonomic rank at the genus level using the *FungalTraits* database for Sutton and Mégantic, respectively[120]. We assigned ASVs to a total of five functional guilds at each of two sites: (i) mutualists (including AMF and ECM fungi), (ii) saprotrophs, (iii) pathogens (including animal, lichen parasites and mycoparasites), (iv) other (including root endophytes and lichenized fungi), and (v) unknown fungi (fungi that could not be assigned to any functional guild) (see Supplementary Data 6, 7). We then calculated the relative abundance of each fungal guild (i.e., the summed number of reads of all ASVs in each guild out of the total number of reads) for each sample.

## Site similarities in root microbial community dominance

The most prevalent AMF ASVs were respectively assigned to the family Glomeraceae in roots and soil, while many mycorrhizal ASVs (62%) could

not be assigned to a family (Supplementary Fig. 3; Supplementary Table 7). For fungi, Sutton and Mégantic showed similar patterns in their predominant fungal classes and families (Supplementary Fig. 4a–d). Specifically, the *Agaricomycetes* class was predominant (S: 43%; M: 38%), closely followed by *Leotiomycetes* (S: 31%; M: 30%) (Supplementary Fig. 4a, b). Among fungal families the *Hyaloscyphaceae* (15%) and *Tricholomataceae* (11%) were most relatively abundant in Sutton (Supplementary Fig. 4c), while it was the *Hyaloscyphaceae* (14%) and *Entolomataceae* (10%) in Mégantic (Supplementary Fig. 4d). For bacteria, the classes *Actinobacteria* (S: 38%; M: 40%) and *Alphaproteobacteria* (S: 19%; M: 17%) were shown to be most relatively abundant at both sites (Supplementary Fig. 4e, f). Finally, the bacterial families *Actinospicaceae* (S: 13%, M: 12%) and *Xanthobacteraceae* (S: 11%, M: 10%) were most dominant at both sites (Supplementary Fig. 4g, h).

## Statistics and reproducibility
All statistical analyses and visualization were conducted using R (version 4.2.1)[115] on the seedlings ($n = 98$) sampled at our two sites (Sutton [S] $n = 51$ / Mégantic [M] $n = 47$)[121,122]. Two samples did not have an entry of seedling height and an entry of annual growth, respectively. The values for these samples were imputed with a principal component analysis using the *missMDA* package[123]. We also performed non-parametric Kruskal-Wallis tests[124] to assess the differences in environmental parameters between sites. We performed non-parametric Kruskal-Wallis tests[124] to assess the differences in environmental parameters between sites. For all analyses described below, each matrix (e.g., root bacteria) contains 98 values, which represent 98 independent sampling units (i.e., seedlings). All our analyses are reproducible from our published raw sequences, datasets, and codes[121,122].

We calculated fungal, bacterial, soil and root AMF richness and Shannon diversity on rarefied matrices using the function *estimate.richness* from the *phyloseq* package[125]. We repeated rarefaction randomly 100 times to compute averaged alpha-diversity (richness and Shannon index) metrics following the steps outlined in Schloss[126]. These averages for fungal, bacterial, root AMF and soil AMF were used in subsequent analyses.

For community composition analyses, we used normalized our data with variance stabilizing transformations (VST) in *DESeq2* to account for uneven sequencing depth[127,128]. We analyzed the variation in microbial community composition (based on Bray-Curtis dissimilarity index) using permutational multivariate analyses of variance (PERMANOVAs)[129,130] with 9999 permutations, performed with the *adonis2* function in *vegan*[128]. To further explore the determinants of microbial community composition, we conducted Principal Coordinates Analysis (PCoA) on ASV tables, extracting site scores from the first and second axes (Multi-Dimensional Scaling [MDS] 1 & 2) for each microbial community[131]. As a preliminary step in building our structural equation models (SEMs), we evaluated the associations between abiotic and biotic drivers with root-associated microbial alpha-diversity (richness, Shannon index) and beta-diversity (MDS1, MDS2) using the *lm* function in R, following the approach used in Laforest-Lapointe et al.[132]. We used the function *plot_model* in *sjPlot* to assess models' fit[133]. To achieve normality in residual distribution, we log-transformed a list of predictors using the function *bestNormalize* (see Supplementary Data 1–5 for details on transformed predictors)[134]. We used $p \leq 0.05$ as the significance threshold.

## Linear and multivariate models
To explore the effects of environmental factors on fungal, bacterial, root, and soil AMF richness and diversity at each site, we focused on predictors that were not or weakly collinear with one or a combination of the considered predictors using variance inflation factor (VIF). In brief, we removed predictors sequentially, starting with those that exhibited the highest VIF (see Supplementary Fig. 5 for removed predictors). The VIFs of the subset were all lower than the recommended cut-off value of 3 in all final models, indicating that multicollinearity did not significantly affect model inference[135]. The final list of predictors for Sutton included elevation, canopy openness, seedling growth, distance to conspecifics, conspecific diameter,

soil chemistry (soil moisture, pH, Ca), AMF root length colonization, vesicle colonization, and total DSE colonization (Supplementary Fig. 5a). For Mégantic, the final set included elevation, canopy openness, seedling growth, distance to conspecifics, conspecific diameter, soil chemistry (soil moisture, pH, Ca, P), AMF arbuscule colonization, vesicle colonization, and total DSE colonization (Supplementary Fig. 5b). The same sets of predictors were used to test the effect of abiotic and biotic factors on root and soil AMF, fungal and bacterial richness, Shannon index, MDS1, MDS2, and community composition.

## Multivariate associations
To evaluate the covariations between root-associated microbial communities, environmental factors, and neighboring plant communities, we calculated RV coefficients with *coeffRV* function of the *FactoMineR* R package[136], a multivariate generalization of the squared Pearson's correlation. RV coefficients quantify the correlation between two matrices with corresponding rows[137,138], giving a single value ranging between 0 (no correlation) and 1 (perfect correlation). RV coefficient significance was assessed using 999 permutations following Josse et al.[139]. We measured the covariation between each pair of the nine following matrices: (1–4) fungal, bacterial, soil and root AMF; (5) aboveground abiotic environment (canopy openness and elevation); (6) soil chemistry (soil moisture, pH, C, N, P, K, Ca, and Mg); (7) conspecific and root microscopy (distance to conspecific, conspecific diameter, AMF root length colonization, arbuscule, vesicle colonization, and total DSE colonization); (8) host characteristics (annual growth, as well as foliar P, K, Ca, and Mg); and (9) the neighboring plant community (Supplementary Fig. 6). We also investigated the degree of association between each pair of root microbial matrices (e.g., root bacteria – root fungi, root fungi – root AMF).

## Structural equation models
We conducted Structural Equation Models (SEM) to investigate the direct and indirect effects of soil chemistry, canopy openness, soil moisture, distance to conspecifics, conspecific diameter, AMF root length, vesicle, arbuscule and DSE colonization, as well as soil- and root-associated microbial communities (soil AMF MDS1, root AMF MDS1, fungal MDS1, and bacterial MDS1) on seedling growth, using the package *PiecewiseSEM*[140]. Models included five exogenous variables: seedling growth, soil AMF MDS1, root AMF MDS1, fungal MDS1, and bacterial MDS1. In examining the indirect impact of the environment on seedling growth through microbial communities, we operated under the assumption that the primary factors influencing microbial communities were those previously identified to have a significant association with microbial communities in linear regressions and PERMANOVAs (Supplementary Fig. 7a, b for Sutton and Mégantic, respectively). We then assessed the overall model fit using direction separation tests (d-sep) based on Fisher's C statistics with models being considered if $p > 0.1$. We simplified our models using a backward stepwise elimination procedure for which we consecutively removed pathways with the highest $p$-value. Endogenous variables were allowed to drop from the model in case effects were not significant ($p > 0.05$). The model with the lowest Akaike information criterion was then selected as the best fit base model.

## Relative abundance of functional guilds and multivariate generalized linear model
To further investigate the significant association that was detected between fungal community composition and seedling growth, we used two approaches. First, we used the *lm* function in R to model seedling annual growth as the response variable, with the e relative abundances of fungal mutualists, saprotrophs, pathogens, other fungi, and unknown fungi as predictor variables. Second, to explore, which fungal taxa can explain a significant association that was detected between fungal community composition and seedling growth, we ran a multivariable generalized linear model (GLM). That is, a GLM was fitted to each of the 20 most abundant fungal families and genera using the *manyglm* function in the *mvabund*

package[141]. In total four models, two for Sutton and two for Mégantic, were fitted using negative binomial probability distribution. VST-transformed ASV tables were used as response variables. To do that, we modeled the 20 most abundant families and genera in Sutton and Mégantic as a function of seedling growth.

## Reporting summary

Further information on research design is available in the Nature Portfolio Reporting Summary linked to this article.

## Data availability

All amplicon sequencing data generated in this study is deposited on the National Center for Biotechnology Information's (NCBI) Sequence Read Archive under BioProject accession number PRJNA1065908 and are available here: www.ncbi.nlm.nih.gov/bioproject/PRJNA1065908. All metadata, taxa, and ASV tables and the numerical source data behind for Fig. 2 are available on Figshare[121].

## Code availability

All scripts for data analysis are available on Figshare[122].

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

## Acknowledgements

We would like to thank Sarah Ishak, Ema Lussier, Philippe Routhier, Rock Ouimet, Mark Vellend, and Sophie Boutin for providing support during fieldwork. We would like to thank Tonia de Bellis for insightful suggestions and discussions during study design and mycorrhizal microscopy. The authors thank Dominique Gravel and Mark Vellend for providing feedback during the project as well as Steven Kembel and again Mark Vellend for providing comments on an earlier version of the manuscript. This research was supported with funding by the Fonds de Recherche du Québec Nature et Technologies (FRQNT; I.L.L. & MF.), the Natural Sciences and Engineering Research Council of Canada (NSERC; I.L.L.), and the Canada Research Chairs program (I.L.L.).

## Author contributions

J.C., I.L.L., and P.-L.C. designed the study. J.C. conducted the sample collection, processing, and molecular work under the supervision of P.-L.C. and I.L.L. J.C., M.F., P.-L.C., and I.L.L. conducted the bioinformatic analyses. M.F. analyzed the data with support from J.C., I.L.L., and F.G.B. J.C. and M.F. wrote the first draft. ILL, MF, F.G.B., and P.-L.C. reviewed the manuscript until the final version.

## Competing interests

The authors declare no competing interests.
