## [Peer Review File · Communications Biology]

Reviewers' comments:

Reviewer #1 (Remarks to the Author):

In the manuscript "The interplay of biotic and abiotic factors shapes tree root endophyte communities and seedling growth", the authors sampled maple seedlings across an elevation gradient at two sites and compared growth with a large number of biotic and abiotic factors. They found that while there is a strong pattern of seedling growth, the large number of parameters that could explain these growth make it difficult to build very strong and convincing evidence for this finding.

I find that the manuscript was very well-written, clearly demonstrating the care that went into it prior to submission. I appreciate the clear language that was used. Despite that, part of the manuscript was difficult to get through, especially the methods section. I suggest that instead of simply restating what was found in the tables, provide a clearer narrative/story for the findings.

In general, I wasn't able to follow a clear central storyline of this manuscript. There were quite a few hypotheses that were being tested, but nothing came through as the main path for readers to follow. That said, there isn't anything "wrong" with what is presented, but I would suggest narrowing down on one or two concepts that the rest of the evidence can be used to support. Due to the descriptive nature of this study, following one or two main concepts can help hone in the message.

The Enemy-Release Hypothesis was invoked but there was no evidence measuring or identifying pathogenic microorganisms in this paper. Since this was one of the major hypotheses used to formulate this paper, I highly suggest that the authors identify those potential pathogens in their amplicon datasets. Popular bioinformatic tools such as FUNGuild can help find those potential pathogens and build some support for this idea.

I was confused throughout the paper because the authors measured both arbuscular mycorrhizal fungi and dark-septate root endophytes, but sometimes only root endophyte was used. Please clarify whether you considered arbuscular mycorrhizal fungi within this endophyte category.

Please provide accession numbers for data repositories.

I have included other comments in the attached PDF for the authors to consider.

Reviewer #2 (Remarks to the Author):

This paper analyzed the environmental factors which affect AMF, fungal and bacterial communities in sugar maple roots and surrounding soil in two elevational gradients in Canada. The sampling is well-designed, and the results are worth publishing. However, the below point should be discussed. In the SEM model, the direction of arrow is from rhizosphere community and other factors to the seedling growth, but in fact, seedling growth must influence the rhizosphere community as well. The author attributed the greater seedling growth in higher elevation to the negative density-dependence hypothesis, but other possibility should also be discussed. In general, the most influential environment to seedling growth is light and water availability. Light condition is only evaluated by canopy openness; however, it is darker under evergreen spruce canopy than deciduous hardwoods even if canopy openness is the same. Models including "neighbor canopy species => light condition => seedling growth => rhizosphere community and soil nutrition" should be considered.

Reviewer #3 (Remarks to the Author):

This manuscript represents a very well-written and scientifically accurate account of extensive study on the biotic and abiotic factors driving sugar maple distribution and endophytic community composition. I congratulate the authors for writing such a well-structured manuscript. However, I

do have one concern regarding the use of the term 'endophyte'- what guarantees do the authors have that the extracted DNA was from endophytic communities, and not from rhizosphere communities attached to the roots? I would thus suggest that the authors change the term "endophytic" to "root-associated". In addition, see below some minor corrections the authors should address:

Line 10: What do you mean by "short marker sequencing"? I do not think that is the correct term – rather use "amplicon sequencing", or "next generation sequencing".

Introduction:

Lines 42-44: missing ref.

Lines 57-74: In my opinion, your introduction is quite extensive and contains information that, while very informative, is not very relevant to the study. An example would be these lines, which highlight points you don't really touch on in your study. Therefore I would consider cutting this section all-together and synthesize it into one or two lines.

Lines 101-103: I think that since you using maple as a model, you could further extrapolate that this study would increase our understanding on tree distribution (not just maple) will shift. I would thus suggest that you state the relevance of your study beyond just the model system you are looking at.

Methods

Lines 195: Please include coverage and size of reads if not already included in table S1.

Lines 197-198: In my opinion, this section describes read processing, rather than bioinformatics analyses (for me these are described in the next section). Thus I would suggest for you to rephrase to "Read processing was performed using dada2...". In addition, please specify the options used when processing the reads , for instance quality scores, truncation values, etc.

Lines 241-243: Regarding the possible biotic and abiotic drivers of beta-diversity, the common method is to perform a constrained analysis such as a CCA or RDA, rather than follow the procedure the authors highlighted. Can the authors justify the choice for methodology?

Lines 245: Please include reference for Kruskal-Wallis tests.

Line 290: Include reference for Fisher's C statistics.

Results:

Lines 382-383: Awkward phrasing – please re-write. In addition, in my opinion these results do not merit describing and discussing due to the lack of significant results.

Discussion:

Lines 385-387: Please include some references for this body of evidence.

Line 387: Delete "Here,".

Lines 417-419: References for this statement?

Lines 470-476: It would be interesting to perform a co-occurrence network to further explore these inter-kingdom correlations.

The Reviewers' comments are shown below in black font while the answers by the authors are in blue font. In addition, the responses to reviewers are numbered (R1, R2, etc.) to facilitate reference to previous queries. The lines referenced to in answers correspond to those in the tracked changed version of the manuscript.

Response to Reviewers:

Reviewer 1

In the manuscript "The interplay of biotic and abiotic factors shapes tree root endophyte communities and seedling growth", the authors sampled maple seedlings across an elevation gradients at two sites and compared growth with a large number of biotic and abiotic factors. They found that while there is a strong pattern of seedling growth, the large number of parameters that could explain these growths make it difficult to build very strong and convincing evidence for this finding. I find that the manuscript was very well-written, clearly demonstrating the care that went into it prior to submission. I appreciate the clear language that was used. Despite that, part of the manuscript was difficult to get through, especially the methods section. I suggest that instead of simply restating what was found in the tables, provide a clearer narrative/story for the findings.

R2: We thank Reviewer 1 for their kind words, for their succinct summary of our work, and for challenging us to sharpen our methods and results sections. We have now improved our methods (see e.g. lines 177 – 179, 270 – 280, 323 – 332, and 340 – 344) and moved some of the methodological descriptions to the *Supporting Information* document, including details on foliar elements quantification and bioinformatics steps, as well as streamlined results sections (see lines 421 – 719, and in Supporting Information at lines 14 – 25 and 37 – 64).

In general, I wasn't able to follow a clear central storyline of this manuscript. There were quite a few hypotheses that were being tested, but nothing came through as the main path for readers to follow. That said, there isn't anything "wrong" with what is presented, but I would suggest narrowing down on one or two concepts that the rest of the evidence can be used to support. Due to the descriptive nature of this study, following one or two main concepts can help home in the message.

R3: Based on the suggestions by Reviewers 1 and 2, we have now refined the level of complexity and detail, improved the structure, and streamlined the flow in the *Introduction*. For example, we shortened the introduction by focusing on two key ecological concepts, the *enemy-release* and *enhanced mutualism* hypotheses, which can help explain the role of microbes in tree establishment and performance at the range limits (see lines 126 – 142) and revised the manuscript to more explicitly state the broader relevance of our study beyond the specific context of sugar maple, for example by removing the paragraph that focused solely on sugar maple (see *Introduction* and Response R27).

The Enemy-Release Hypothesis was invoked but there was no evidence measuring or identifying pathogenic microorganisms in this paper. Since this was one of the major hypotheses used to formulate this paper, I highly suggest that the authors identify those potential pathogens in their amplicon datasets. Popular bioinformatic tools such as FUNGuild can help find those potential pathogens and build some support for this idea.

R4: Thank you for the great suggestion. We have now assigned functional guilds to our fungal data. To do that, we manually assigned functional traits to fungal ASVs based on their taxonomic rank at the genus level using the *FungalTraits* database¹. We assigned ASVs to a total of five functional guilds: i) mutualists (including AMF and EMF fungi), ii) saprotrophs, iii) pathogens (including animal, lichen parasites and mycoparasites), iv) other (including root endophytes and lichenized fungi), and v) unknown fungi (fungi that could not be assigned to any functional guild). This information is now included in the supplement (see Supplement Data 1 and 2) as well as mentioned in the *Methods* section (lines 281 – 289 and 400 – 410), *Results* (709 – 719) and *Discussion* (872 – 884) sections.

I was confused throughout the paper because the authors measured both arbuscular mycorrhizal fungi and dark-septate root endophytes, but sometimes only root endophyte was used. Please clarify whether you considered arbuscular mycorrhizal fungi within this endophyte category.

R5: We have now taken upon the suggestion of Reviewers 1 and 3 and talk about root-associated communities rather than endophytes throughout the paper (for example, see lines 13, 16, 23, 53, and 143). We have also modified the title of the paper to reflect the changes made.

Please provide accession numbers for data repositories.

R6: We have done so at line 930.

I have included other comments in the attached PDF for the authors to consider. Please provide pH measurement for these soils (Line 104).

R7: We have now provided this information at lines 177 – 179.

Lines 140 and 165: This data was not presented in the results nor discussion (at least as far as I can see). If it's not important, please remove it or briefly address it in the results.

R8: Indeed, foliar elements concentrations have been used only in one part of the analyses, in which we correlated host characteristics with microbial community data. As it plays a very minor role in this work and to follow the advice of simplifying the methods/results section, we have moved this text to the *Supplementary Information* in the section *Foliar elements quantification* spanning lines 14 to 25.

Lines 234-235: This is a logical fallacy. Just because everyone is using it doesn't make it the proper choice. I recommend providing a more sound reason, or remove this part of the sentence.

R9: We agree and have now removed this subsentence.

Line 241: Please define "root endophyte". I read through the whole discussion but still was confused whether arbuscular mycorrhizal fungi were included as endophytes. Typically in the literature they are not written as endophytes, but of course, they are.

R10: As stated in Responses 5 and 24, we have now avoided the term “root endophyte” and instead use term “root-associated microbial communities” throughout the manuscript, including the title of the paper.

Line 314: I believe that this is amplicon data. Please be more clear here as you also have colonization data.

R11: You are right that AMF root and soil multi-dimensional scaling [MDS1 and MDS2] were calculated using ASV tables. We have now clarified this at Line 339.

Line 381: Please provide reasoning for analyzing this at the family level. There isn't much one can learn at that level. I would suggest analyzing at the genus level where some ecosystem process can be attributed.

R12: Following the reviewer comment we have now also analyzed this data at the genus-level and have added the results of the analysis to the *Method* and *Results* sections at lines 281 – 289 and 400 – 410 and 709 – 719, as well as in Table S11. Additionally, as stated in Response 4, we have assigned functional guilds to the fungal ASVs using *FungalTraits* database to identify potential pathogens in our data and ran linear models to explore whether the relative abundance of pathogens changed along seedling annual growth. We did not find a significant association between the relative abundance of fungal pathogens and seedling annual growth thus we discussed this finding at lines 872 – 884.

Line 392: This is odd here. Although neighbouring tree communities have strong correlations to certain microbial community measurements, it did not appear in the model. Please explain why these tree communities would be important here.

R13: We have now removed the confusing sentence to emphasize that we meant distance to conspecific trees (Lines 730 – 732).

Line 421: Are these associations positive or negatively correlated? This could help explain some of the discussion in this paragraph.

R14: The findings of our study revealed a nuanced relationship between dark septate endophytes (DSE) and arbuscular mycorrhizal fungi (AMF) in both root and soil. Specifically, we observed a negative association of DSE with root AMF (MDS2 axis), contrasting with a significant positive association with soil AMF (MDS2 axis), as detailed in Table 1. However, interpreting these associations is challenging due to the nature of our data, which are site scores derived from the first and second axes of a Multi-Dimensional Scaling (MDS 1 & 2) analysis. These axes represent broad patterns in AMF soil and root community composition across samples and do not convey explicit information on species associations. To address this complexity and potential ambiguity, we discuss the diverse ecological roles of DSE, ranging from mutualistic to pathogenic, and their varying interactions with AMF communities (see lines 779 – 799). We have now also clarified our discussion on the interactions between DSE and AMF at lines 774 – 779.

Line 433: Most elevation studies have a much stronger gradient (e.g. 0-1000m) than your study. Could this help to explain the pattern here?

R15: We thank the reviewer for the insightful comment regarding the scale of elevation gradients in our study. We acknowledge that most elevation studies typically investigate more pronounced gradients (e.g., 0-1000m), which is a significant aspect to consider. However, even if the gradients reported in our study are smaller in distance, they show important changes in plant community composition and forest ecosystem. In addition, there seems to be no agreement in literature on whether changes in root and soil microbial richness and diversity follow a universal pattern. Previous studies have reported increases in diversity at higher elevations, declines, hump-shaped patterns, or no change. To address the reviewer's comment, we now introduce this argument at lines 804 – 808.

Line 484 to 489: This is too weak of an evidence to discuss here. There are tools available now, such as FUNGuild, that allows you to pick out potential pathogens from your dataset. Identifying pathogens across the range would be important for your hypothesis.

R16: As stated in Response 4 and 12, we have taken upon the suggestion of the reviewer and assigned fungal functional guilds using *FungalTraits* database. We have also conducted additional statistical analyses to explore whether fungal functional guilds, for example mutualists, saprotrophs, and pathogens, differed in their relative abundance with seedling annual growth (see lines 281 – 289, 400 – 410, 709 – 719 and 872 – 884).

Line 493: I am not able to make a clear connection between the correlation between fungal and bacterial communities (which metric?) and how that is connected to inter-kingdom interactions? Please provide more clear reasoning to make this jump.

R17: We have clarified this part by making it clear that we discuss the results from covariation analysis, where we explored the correlation between pairs of microbial matrices using a multivariate generalization of the squared Pearson's correlation (lines 892).

Line 504: There is a strong connection between soil pH and calcium. Has a correlation analysis been done to make sure that they are not co-correlated?

R18: Prior to our analysis, we carefully explored the correlations among our variables (for example Figure S2) and removed predictors sequentially, starting with those that exhibited the highest variance inflation factor (VIF) as stated at lines 350 – 356 in the *Statistical analysis* section. In all final models, the VIFs of the predictors were all lower than the recommended cut-off value of 3, indicating that multicollinearity did not significantly affect model inference, including the collinearity between soil pH and Ca.

Line 508: Correlated with. Since this is a descriptive dataset, it's not strong enough to think of them as determinants here.

R19: We fully agree that our choice of words was not always compatible with the observational nature of our study. We have now carefully reconsidered our wordings and our strong statements (see lines 19 and 916).

Line 514: Please cite accession numbers for both repositories.

R20: This is now added.

Reviewer 2

This paper analyzed the environmental factors which affect AMF, fungal and bacterial communities in sugar maple roots and surrounding soil in two elevational gradients in Canada. The sampling is well-designed, and the results are worth publishing. However, the below point should be discussed.

R21: We thank Reviewer 2 for these positive words.

In the SEM model, the direction of arrow is from rhizosphere community and other factors to the seedling growth, but in fact, seedling growth must influence the rhizosphere community as well. The author attributed the greater seedling growth in higher elevation to the negative density-dependence hypothesis, but other possibility should also be discussed.

R22: We agree that the observed relationship between tree seedling growth and root-associated microbial communities can be bidirectional. However, the novelty of our finding hinges not on the establishment of a unidirectional causal link between root-associated microbial community and seedling growth, but rather on the very existence of strong correlations with root fungi rather than with bacteria or mycorrhizae. Yet, to address the reviewer's comment, in the *Discussion* (lines 901 – 903) we stated explicitly that trees can also affect soil microbial communities.

We also agree that other scenarios exist that can explain greater seedling growth at higher elevation. Following the reviewer's comment, we have now added that this pattern can also be explained by inherited properties of fungal communities (e.g., networks, inter-taxa, and inter-kingdom interactions), lower herbivore pressure or variation in microclimate at lines 884 – 889 in the *Discussion*.

In general, the most influential environment to seedling growth is light and water availability. Light condition is only evaluated by canopy openness; however, it is darker under evergreen spruce canopy than deciduous hardwoods even if canopy openness is the same. Models including "neighbor canopy species => light condition => seedling growth => rhizosphere community and soil nutrition" should be considered.

R23: We thank Reviewer 2 for the insightful comments regarding the importance of light and water availability in seedling growth and the suggestion to consider the influence of neighbor canopy species on light conditions. We agree that the type of canopy cover, particularly the distinction between evergreen spruce and deciduous hardwoods, can significantly impact the light availability for tree seedlings, even with similar canopy openness levels. Recognizing this, we have now discussed in our manuscript the potential impact of neighboring canopy species on light conditions and subsequently on seedling growth (see lines 862 – 871).

Reviewer 3

This manuscript represents a very well-written and scientifically accurate account of extensive study on the biotic and abiotic factors driving sugar maple distribution and endophytic community composition. I congratulate the authors for writing such a well-structured manuscript. However, I do have one concern regarding the use of the term 'endophyte'- what guarantees do the authors have that the extracted DNA was from endophytic communities, and not from rhizosphere communities attached to the roots? I would thus suggest that the authors change the term "endophytic" to "root-associated". In addition, see below some minor corrections the authors should address:

R24: We thank Reviewer 3 for their kind words and helpful comments to improve the manuscript.

The process to remove as many ectophytes as possible is mentioned at line 213 to 215: "*Roots were cleaned using distilled water to remove residual attached soil particles and organic matter.*" Yet, we agree that we cannot demonstrate that we removed all rhizosphere microorganisms. Addressing at the same time a suggestion from Reviewer 1 on AMF not being defined as endophytes, we have thus transitioned from using the term *endophytes* to *root-associated microbial communities* throughout the manuscript (for example, see lines 13, 16, 23, 52 and 143) and the new version of the title of the paper.

Line 10: What do you mean by "short marker sequencing"? I do not think that is the correct term – rather use "amplicon sequencing", or "next generation sequencing".

R25: Thanks for pointing to this glitch, we have now corrected it.

Introduction. Lines 42-44: missing ref.

R26: The references have been added.

Lines 57-74: In my opinion, your introduction is quite extensive and contains information that, while very informative, is not very relevant to the study. An example would be these lines, which highlight points you don't really touch on in your study. Therefore I would consider cutting this section all-together and synthesize it into one or two lines.

R27: As pointed out in Responses 2, 3, and 28, we have taken utmost care to streamline our Introduction. To address the reviewers' comments, we have now further shortened the introduction, specifically focusing on the *enemy-release* and *enhanced mutualism hypotheses* (see lines 126 to 142). We believe that these key ecological concepts are crucial for our study as they provide a theoretical framework for understanding the role of microbes in tree establishment and performance at range limits. These hypotheses are not only central to our research question but also guide our interpretation of the results, especially in the context of root microbial community dynamics and seedling growth.

Lines 101-103: I think that since you using maple as a model, you could further extrapolate that this study would increase our understanding on tree distribution (not just maple) will shift. I would thus suggest that you state the relevance of your study beyond just the model system you are looking at.

R28: We thank Reviewer 3 for this insightful comment, we have modified the Introduction to be more careful in stating the relevance of our study beyond one single tree species (e.g., sugar maple) distribution. To do that, we have now removed the paragraph on sugar maple and modified the last paragraph of the Introduction (see lines 143 to 149).

Methods

Lines 195: Please include coverage and size of reads if not already included in table S1.

R29: We have included this information in Tables S2 and S3.

Lines 197-198: In my opinion, this section describes read processing, rather than bioinformatics analyses (for me these are described in the next section). Thus, I would suggest for you to rephrase to “Read processing was performed using dada2...”. In addition, please specify the options used when processing the reads, for instance quality scores, truncation values, etc.

R30: We agree with the reviewer and have made the suggested modification / additions in the *Methods* section and have moved it to the *Supporting Information* document (see lines 37 to 48).

Lines 241-243: Regarding the possible biotic and abiotic drivers of beta-diversity, the common method is to perform a constrained analysis such as a CCA or RDA, rather than follow the procedure the authors highlighted. Can the authors justify the choice for methodology?

R31: We agree with the reviewer that many studies on microbial community dynamics employ multivariate analyses including PCoAs, CCAs, and RDAs to explore the relative importance of environmental drivers on community structure. When exploring the variation in root microbial community composition (beta-diversity) we are relying on permutational multivariate analyses of variance (PERMANOVAs), which rely essentially on canonical ordinations such as CCAs and RDAs.

Regarding the modeling of site scores from MDS1 and MDS2, we have employed linear models as an initial step towards developing our structural equation models (SEMs). This approach is consistent with the methodology adopted in prior research, such as the study by Laforest-Lapointe et al. (2017, *Nature*)², which utilized uses linear regressions to initially test the relationship between microbial beta- and alpha-diversity and abiotic factors. These relationships then inform the construction of SEMs, which are based on the outcomes of the most informative linear models.

We believe that this combined approach of linear models for preliminary steps, PERMANOVA for multivariate analysis, and SEMs provide a balanced and effective methodological framework for our research objectives. Furthermore, from a technical perspective, the PERMANOVA we used relies on distance-based RDA, which itself relies on multiple linear regression. The mathematical details showing the links between PERMANOVA, RDA, db-RDA

and multiple linear regression are presented in section 11.1 of Legendre and Legendre (2012)³. We have now clarified our rationale at lines 340 – 344 and 388 – 392.

Lines 245: Please include reference for Kruskal-Wallis tests.

R32: The reference has been added.

Line 290: Include reference for Fisher's C statistics.

R33: The reference has been added.

Results:

Lines 382-383: Awkward phrasing – please re-write. In addition, in my opinion these results do not merit describing and discussing due to the lack of significant results.

R34: We have now modified this part since we added results on the differences in the relative abundance of fungal functional guilds along with the taxa along seedling growth (709 – 719).

Discussion:

Lines 385-387: Please include some references for this body of evidence.

R35: We have added the adequate references.

Line 387: Delete "Here,".

R36: Done.

Lines 417-419: References for this statement?

R37: We have now added reference along with this statement on Line 773.

Lines 470-476: It would be interesting to perform a co-occurrence network to further explore these inter-kingdom correlations.

R38: It would indeed be interesting to study the inter-kingdom correlations with a co-occurrence network, however, while we initially considered this approach, we decided to follow alternative analytical routes because it is difficult to infer what such correlations would represent. Blanchet et al. (2020) of recent works (see Blanchet et al. 2020 for example)⁴ have shown that showing that there are many reasons why co-occurrence may appear. For example, the correlations highlighted by a co-occurrence network could be associated to direct interactions but also indirect ones or a common environmental forcing, and there is often no way to clearly decide which situation is the one that drives the system. As our interest lies in understanding the impact of biotic and abiotic factors on microbial communities, using a tool (e.g. co-occurrence network) that is well adapted to describe inter-kingdom correlations, but not efficient at making inferences on the cause of these correlations, bring us halfway towards our end goal. In this respect, we chose to approach the question from another angle (the one presented in the manuscript).

However, to address the reviewer's comment, we have now clearly stated that experimental studies are needed to validate the nature of these associations, e.g. to distinguish between

co-occurrence that are the results of an environmental forcing and true ecological interactions, such as competition or mutualism among microbial taxa at lines 903 – 906, 884 – 887 and lines 797 – 799.

Moreover, we may soon answer this question ourselves, since the follow-up from this manuscript is an experimental study in greenhouses with live soils from the two gradients, which among other things, explores the nature of microbial inter-taxa interactions as well as their individual and joint effect on seedling fitness. This study will contribute valuable insights to our current understanding and fill in some of the gaps in the multi-kingdom interactions of microbes by going from observation to experimentation.

References:

1. Pölme, S. *et al.* FungalTraits: a user-friendly traits database of fungi and fungus-like stramenopiles. *Fungal Diversity* **105**, 1–16 (2020).
2. Laforest-Lapointe, I., Paquette, A., Messier, C. & Kembel, S. W. Leaf bacterial diversity mediates plant diversity and ecosystem function relationships. *Nature* **546**, 145–147 (2017).
3. Legendre, P. & Legendre, L. Numerical ecology. in *Numerical ecology* vol. 24 (Elsevier Science, Amsterdam, The Netherlands, 2012).
4. Blanchet, F. G., Cazelles, K. & Gravel, D. Co-occurrence is not evidence of ecological interactions. *Ecology Letters* **23**, 1050–1063 (2020).

REVIEWERS' COMMENTS:

Reviewer #2 (Remarks to the Author):

Thank you for revising the manuscript. Now the manuscript seems ready to be published.

Reviewer #3 (Remarks to the Author):

In the revised version of this manuscript, the authors have address all of my concerns. I congratulate the authors again for the well-executed and well-written study.